# Negative Data Augmentation

**Abhishek Sinha**[1]* **Kumar Ayush**[1]* **Jiaming Song**[1]* **Burak Uzkent**[1] **Hongxia Jin**[2]

**Stefano Ermon**[1]

Department of Computer Science[1]
Stanford University
{a7b23, kayush, tsong, buzkent, ermon}@stanford.edu

Samsung Research America[2]

## Abstract

Data augmentation is often used to enlarge datasets with synthetic samples generated in accordance with the underlying data distribution. To enable a wider range of augmentations, we explore *negative* data augmentation strategies (NDA) that intentionally create out-of-distribution samples. We show that such negative out-of-distribution samples provide information on the support of the data distribution, and can be leveraged for generative modeling and representation learning. We introduce a new GAN training objective where we use NDA as an additional source of synthetic data for the discriminator. We prove that under suitable conditions, optimizing the resulting objective still recovers the true data distribution but can directly bias the generator towards avoiding samples that lack the desired structure. Empirically, models trained with our method achieve improved conditional/unconditional image generation along with improved anomaly detection capabilities. Further, we incorporate the same negative data augmentation strategy in a contrastive learning framework for self-supervised representation learning on images and videos, achieving improved performance on downstream image classification, object detection, and action recognition tasks. These results suggest that prior knowledge on what does not constitute valid data is an effective form of weak supervision across a range of unsupervised learning tasks.

## 1 Introduction

Data augmentation strategies for synthesizing new data in a way that is consistent with an underlying task are extremely effective in both supervised and unsupervised learning (Oord et al., 2018; Zhang et al., 2016; Noroozi & Favaro, 2016; Asano et al., 2019). Because they operate at the level of samples, they can be combined with most learning algorithms. They allow for the incorporation of prior knowledge (inductive bias) about properties of typical samples from the underlying data distribution (Jaiswal et al., 2018; Antoniou et al., 2017), e.g., by leveraging invariances to produce additional "positive" examples of how a task should be solved.

To enable users to specify an even wider range of inductive biases, we propose to leverage an alternative and complementary source of prior knowledge that specifies how a task should *not* be solved. We formalize this intuition by assuming access to a way of generating samples that are guaranteed to be out-of-support for the data distribution, which we call a *Negative Data Augmentation* (NDA). Intuitively, negative out-of-distribution (OOD) samples can be leveraged as a useful inductive bias because they provide information about the support of the data distribution to be learned by the model. For example, in a density estimation problem we can bias the model to avoid putting any probability mass in regions which we know a-priori should have zero probability. This can be an effective prior if the negative samples cover a sufficiently large area. The best NDA candidates are ones that expose common pitfalls of existing models, such as prioritizing local structure over global

---

*Equal Contribution

structure (Geirhos et al., 2018); this motivates us to consider known transformations from the literature that intentionally destroy the spatial coherence of an image (Noroozi & Favaro, 2016; DeVries & Taylor, 2017; Yun et al., 2019), such as Jigsaw transforms.

Building on this intuition, we introduce a new GAN training objective where we use NDA as an additional source of fake data for the discriminator as shown in Fig. 1. Theoretically, we can show that if the NDA assumption is valid, optimizing this objective will still recover the data distribution in the limit of infinite data. However, in the finite data regime, there is a need to generalize beyond the empirical distribution (Zhao et al., 2018). By explicitly providing the discriminator with samples we want to avoid, we are able to bias the generator towards avoiding undesirable samples thus improving generation quality.

Furthermore, we propose a way of leveraging NDA for unsupervised representation learning. We propose a new contrastive predictive coding (He et al., 2019; Han et al., 2019) (CPC) objective that encourages the distribution of representations corresponding to in-support data to become disjoint from that of NDA data. Empirically, we show that applying NDA with our proposed transformations (e.g., forcing the representation of normal and jigsaw images to be disjoint) improves performance in downstream tasks.

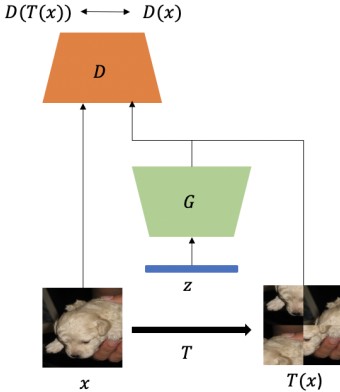

Figure 1: Negative Data Augmentation for GANs.

With appropriately chosen NDA strategies, we obtain superior empirical performance on a variety of tasks, with almost no cost in computation. For generative modeling, models trained with NDA achieve better image generation, image translation and anomaly detection performance compared with the same model trained without NDA. Similar gains are observed on representation learning for images and videos over downstream tasks such as image classification, object detection and action recognition. These results suggest that NDA has much potential to improve a variety of self-supervised learning techniques.

## 2 NEGATIVE DATA AUGMENTATION

The input to most learning algorithms is a dataset of samples from an underlying data distribution $p_{data}$. While $p_{data}$ is unknown, learning algorithms always rely on prior knowledge about its properties (inductive biases (Wolpert & Macready, 1997)), e.g., by using specific functional forms such as neural networks. Similarly, data augmentation strategies exploit known invariances of $p_{data}$, such as the conditional label distribution being invariant to semantic-preserving transformations.

While typical data augmentation strategies exploit prior knowledge about what is in support of $p_{\text{data}}$, in this paper, we propose to exploit prior knowledge about what is *not* in the support of $p_{data}$. This information is often available for common data modalities (e.g., natural images and videos) and is under-exploited by existing approaches. Specifically, we assume: (1) there exists an alternative distribution $\overline{p}$ such that its support is disjoint from that of $p_{data}$; and (2) access to a procedure to efficiently sample from $\overline{p}$. We emphasize $\overline{p}$ need not be explicitly defined (e.g., through an explicit density) – it may be implicitly defined by a dataset or by a procedure that transforms samples from $p_{\text{data}}$ into ones from $\overline{p}$ by suitably altering their structure.

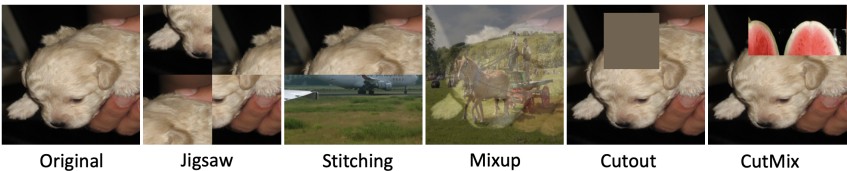

Figure 2: Negative augmentations produce out-of-distribution samples lacking the typical structure of natural images; these negative samples can be used to inform a model on what it should *not* learn.

Analogous to typical data augmentations, NDA strategies are by definition domain and task specific. In this paper, we focus on natural images and videos, and leave the application to other domains (such as natural language processing) as future work. How do we select a good NDA strategy? According to the manifold hypothesis (Fefferman et al., 2016), natural images lie on low-dimensional manifolds: $p_{data}$ is supported on a low-dimensional manifold of the ambient (pixel) space. This suggests that many negative data augmentation strategies exist. Indeed, sampling random noise is in most cases a valid NDA. However, while this prior is generic, it is not very informative, and this NDA will likely be ineffective for most learning problems. Intuitively, NDA is informative if its support is close (in a suitable metric) to that of $p_{data}$, while being disjoint. These negative samples will provide information on the "boundary" of the support of $p_{data}$, which we will show is helpful in several learning problems. In most of our tasks, the images are processed by convolutional neural networks (CNNs) that are good at processing local features but not necessarily global features (Geirhos et al., 2018). Therefore, we may consider NDA examples to be ones that preserve local features ("informative") and break global features, so that it forces the CNNs to learn global features (by realizing NDAs are different from real data).

Leveraging this intuition, we show several image transformations from the literature that can be viewed as generic NDAs over natural images in Figure 2, that we will use for generative modeling and representation learning in the following sections. Details about these transformations can be found in Appendix B.

# 3 NDA FOR GENERATIVE ADVERSARIAL NETWORKS

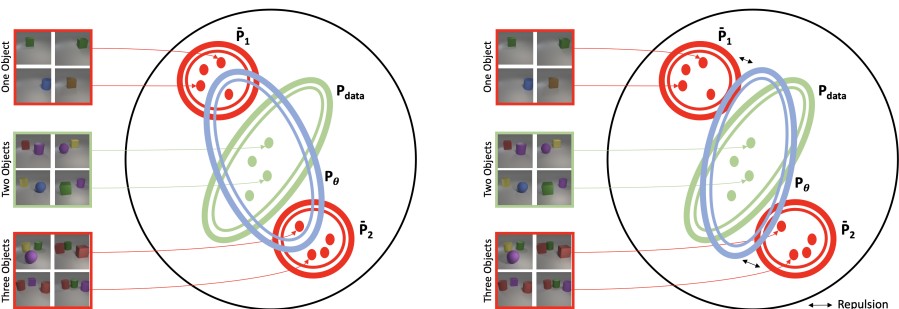

Figure 3: Schematic overview of our NDA framework. **Left**: In the absence of NDA, the support of a generative model $P_\theta$ (blue oval) learned from samples (green dots) may "over-generalize" and include samples from $\overline{P_1}$ or $\overline{P_2}$. **Right**: With NDA, the learned distribution $P_\theta$ becomes disjoint from NDA distributions $\overline{P_1}$ and $\overline{P_2}$, thus pushing $P_\theta$ closer to the true data distribution $p_{data}$ (green oval). As long as the prior is consistent, i.e. the supports of $\overline{P_1}$ and $\overline{P_2}$ are truly disjoint from $p_{data}$, the best fit distribution in the infinite data regime does not change.

In GANs, we are interested in learning a generative model $G_\theta$ from samples drawn from some data distribution $p_{\text{data}}$ (Goodfellow et al., 2014). GANs use a binary classifier, the so-called discriminator $D_\phi$, to distinguish real data from generated (fake) samples. The generator $G_\theta$ is trained via the following mini-max objective that performs variational Jensen-Shannon divergence minimization:

$$\min_{G_\theta \in \mathcal{P}(\mathcal{X})} \max_{D_\phi} L_{\text{JS}}(G_\theta, D_\phi) \quad \text{where} \tag{1}$$

$$L_{\text{JS}}(G_\theta, D_\phi) = \mathbb{E}_{\boldsymbol{x} \sim p_{\text{data}}} \left[ \log(D_\phi(\boldsymbol{x})) \right] + \mathbb{E}_{\boldsymbol{x} \sim G_\theta} \left[ \log(1 - D_\phi(\boldsymbol{x})) \right] \tag{2}$$

This is a special case to the more general variational $f$-divergence minimization objective (Nowozin et al., 2016). The optimal $D_\phi$ for any $G_\theta$ is $(p_{\text{data}}/G_\theta)/(1 + p_{\text{data}}/G_\theta)$, so the discriminator can serve as a density ratio estimator between $p_{\text{data}}$ and $G_\theta$.

With sufficiently expressive models and infinite capacity, $G_\theta$ will match $p_{\text{data}}$. In practice, however, we have access to finite datasets and limited model capacity. This means that the generator needs to generalize beyond the empirical distribution, which is challenging because the number of possible discrete distributions scale *doubly exponentially* w.r.t. to the data dimension. Hence, as studied in (Zhao et al., 2018), the role of the inductive bias is critical. For example, Zhao et al. (2018) report

that when trained on images containing 2 objects only, GANs and other generative models can sometimes "generalize" by generating images with 1 or 3 objects (which were never seen in the training set). The generalization behavior – which may or may not be desirable – is determined by factors such as network architectures, hyperparameters, etc., and is difficult to characterize analytically.

Here we propose to bias the learning process by directly specifying what the generator should *not* generate through NDA. We consider an adversarial game based on the following objective:

$$\min_{G_\theta \in \mathcal{P}(\mathcal{X})} \max_{D_\phi} L_{\mathrm{JS}}(\lambda G_\theta + (1-\lambda)\overline{P}, D_\phi) \tag{3}$$

where the negative samples are generated from a mixture of $G_\theta$ (the generator distribution) and $\overline{P}$ (the NDA distribution); the mixture weights are controlled by the hyperparameter $\lambda$. Intuitively, this can help addresses the above "over-generalization" issue, as we can directly provide supervision on what should not be generated and thus guide the support of $G_\theta$ (see Figure 3) . For instance, in the object count example above, we can empirically prevent the model from generating images with an undesired number of objects (see Appendix Section A for experimental results on this task).

In addition, the introduction of NDA samples will not affect the solution of the original GAN objective in the limit. In the following theorem, we show that given infinite training data and infinite capacity discriminators and generators, using NDA will not affect the optimal solution to the generator, *i.e.* the generator will still recover the true data distribution.

**Theorem 1.** *Let $\overline{P} \in \mathcal{P}(\mathcal{X})$ be any distribution over $\mathcal{X}$ with disjoint support than $p_{\mathrm{data}}$, i.e., such that $\mathrm{supp}(p_{\mathrm{data}}) \cap \mathrm{supp}(\overline{P}) = \varnothing$. Let $D_\phi : \mathcal{X} \to \mathbb{R}$ be the set of all discriminators over $\mathcal{X}$, $f : \mathbb{R}_{\geq 0} \to \mathbb{R}$ be a convex, semi-continuous function such that $f(1) = 0$, $f^\star$ be the convex conjugate of $f$, $\overline{f}'$ its derivative, and $G_\theta$ be a distribution with sample space $\mathcal{X}$. Then $\forall \lambda \in (0, 1]$, we have:*

$$\arg\min_{G_\theta \in \mathcal{P}(\mathcal{X})} \max_{D_\phi:\mathcal{X}\to\mathbb{R}} L_f(G_\theta, D_\phi) = \arg\min_{G_\theta \in \mathcal{P}(\mathcal{X})} \max_{D_\phi:\mathcal{X}\to\mathbb{R}} L_f(\lambda G_\theta + (1-\lambda)\overline{P}, D_\phi) = p_{\mathrm{data}} \tag{4}$$

*where $L_f(Q, D_\phi) = \mathbb{E}_{\boldsymbol{x}\sim p_{\mathrm{data}}}[D_\phi(\boldsymbol{x})] - \mathbb{E}_{\boldsymbol{x}\sim Q}[f^\star(D_\phi(\boldsymbol{x}))]$ is the objective for $f$-GAN (Nowozin et al., 2016). However, the optimal discriminators are different for the two objectives:*

$$\arg\max_{D_\phi:\mathcal{X}\to\mathbb{R}} L_f(G_\theta, D_\phi) = f'(p_{\mathrm{data}}/G_\theta) \tag{5}$$

$$\arg\max_{D_\phi:\mathcal{X}\to\mathbb{R}} L_f(\lambda G_\theta + (1-\lambda)\overline{P}, D_\phi) = f'(p_{\mathrm{data}}/(\lambda G_\theta + (1-\lambda)\overline{P})) \tag{6}$$

*Proof.* See Appendix C. □

The above theorem shows that in the limit of infinite data and computation, adding NDA changes the optimal discriminator solution but not the optimal generator. In practice, when dealing with finite data, existing regularization techniques such as weight decay and spectral normalization (Miyato et al., 2018) allow potentially many solutions that achieve the same objective value. The introduction of NDA samples allows us to filter out certain solutions by providing additional inductive bias through OOD samples. In fact, the optimal discriminator will reflect the density ratio between $p_{\mathrm{data}}$ and $\lambda G_\theta + (1-\lambda)\overline{P}$ (see Eq.(6)), and its values will be higher for samples from $p_{\mathrm{data}}$ compared to those from $\overline{P}$. As we will show in Section 5, a discriminator trained with this objective and suitable NDA performs better than relevant baselines for other downstream tasks such as anomaly detection.

## 4 NDA FOR CONSTRASTIVE REPRESENTATION LEARNING

Using a classifier to estimate a density ratio is useful not only for estimating $f$-divergences (as in the previous section) but also for estimating mutual information between two random variables. In representation learning, mutual information (MI) maximization is often employed to learn compact yet useful representations of the data, allowing one to perform downstream tasks efficiently (Tishby & Zaslavsky, 2015; Nguyen et al., 2008; Poole et al., 2019b; Oord et al., 2018). Here, we show that NDA samples are also beneficial for representation learning.

In contrastive representation learning (such as CPC (Oord et al., 2018)), the goal is to learn a mapping $h_\theta(\boldsymbol{x}) : \mathcal{X} \to \mathcal{P}(\mathcal{Z})$ that maps a datapoint $\boldsymbol{x}$ to some distribution over the representation space

$\mathcal{Z}$; once the network $h_\theta$ is learned, representations are obtained by sampling from $\boldsymbol{z} \sim h_\theta(\boldsymbol{x})$. CPC *maximizes* the following objective:

$$I_{\text{CPC}}(h_\theta, g_\phi) := \mathbb{E}_{\boldsymbol{x} \sim p_{\text{data}}(\boldsymbol{x}), \boldsymbol{z} \sim h_\theta(\boldsymbol{x}), \widehat{\boldsymbol{z}}_i \sim p_\theta(\boldsymbol{z})} \left[ \log \frac{n g_\phi(\boldsymbol{x}, \boldsymbol{z})}{g_\phi(\boldsymbol{x}, \boldsymbol{z}) + \sum_{j=1}^{n-1} g_\phi(\boldsymbol{x}, \widehat{\boldsymbol{z}}_j)} \right] \quad (7)$$

where $p_\theta(\boldsymbol{z}) = \int h_\theta(\boldsymbol{z}|\boldsymbol{x}) p_{\text{data}}(\boldsymbol{x}) \mathrm{d}\boldsymbol{x}$ is the marginal distribution of the representations associated with $p_{\text{data}}$. Intuitively, the CPC objective involves an $n$-class classification problem where $g_\phi$ attempts to identify a matching pair (i.e. $(\boldsymbol{x}, \boldsymbol{z})$) sampled from the joint distribution from the $(n-1)$ non-matching pairs (i.e. $(\boldsymbol{x}, \widehat{\boldsymbol{z}}_j)$) sampled from the product of marginals distribution. Note that $g_\phi$ plays the role of a discriminator/critic, and is implicitly estimating a density ratio. As $n \to \infty$, the optimal $g_\phi$ corresponds to an un-normalized density ratio between the joint distribution and the product of marginals, and the CPC objective matches its upper bound which is the mutual information between $X$ and $Z$ (Poole et al., 2019a; Song & Ermon, 2019). However, this objective is no longer able to control the representations for data that are out of support of $p_{\text{data}}$, so there is a risk that the representations are similar between $p_{\text{data}}$ samples and out-of-distribution ones.

To mitigate this issue, we propose to use NDA in the CPC objective, where we additionally introduce a batch of NDA samples, for each positive sample:

$$\overline{I_{\text{CPC}}}(h_\theta, g_\phi) := \mathbb{E} \left[ \log \frac{(n+m) g_\phi(\boldsymbol{x}, \boldsymbol{z})}{g_\phi(\boldsymbol{x}, \boldsymbol{z}) + \sum_{j=1}^{n-1} g_\phi(\boldsymbol{x}, \widehat{\boldsymbol{z}}_j) + \sum_{k=1}^{m} g_\phi(\boldsymbol{x}, \overline{\boldsymbol{z}}_k)} \right] \quad (8)$$

where the expectation is taken over $\boldsymbol{x} \sim p_{\text{data}}(\boldsymbol{x}), \boldsymbol{z} \sim h_\theta(\boldsymbol{x}), \widehat{\boldsymbol{z}}_i \sim p_\theta(\boldsymbol{z}), \overline{\boldsymbol{x}}_k \sim \overline{p}$ (NDA distribution), $\overline{\boldsymbol{z}}_k \sim h_\theta(\overline{\boldsymbol{x}}_k)$ for all $k \in [m]$. Here, the behavior of $h_\theta(\boldsymbol{x})$ when $\boldsymbol{x}$ is NDA is optimized explicitly, allowing us to impose additional constraints to the NDA representations. This corresponds to a more challenging classification problem (compared to basic CPC) that encourages learning more informative representations. In the following theorem, we show that the proposed objective encourages the representations for NDA samples to become disjoint from the representations for $p_{\text{data}}$ samples, *i.e.* NDA samples and $p_{\text{data}}$ samples do not map to the same representation.

**Theorem 2.** *(Informal) The optimal solution to $h_\theta$ in the NDA-CPC objective maps the representations of data samples and NDA samples to disjoint regions.*

*Proof.* See Appendix D for a detailed statement and proof. □

## 5 NDA-GAN Experiments

In this section we report experiments with different types of NDA for image generation. Additional details about the network architectures and hyperparameters can be found in Appendix K.

**Unconditional Image Generation.** We conduct experiments on various datasets using the BigGAN architecture (Brock et al., 2018) for unconditional image generation[1]. We first explore various image transformations from the literature to evaluate which ones are effective as NDA. For each transformation, we evaluate its performance as NDA (training as in Eq. 3) and as a traditional data augmentation strategy, where we enlarge the training set by applying the transformation to real images (denoted PDA for positive data augmentation). Table 1 shows the FID scores for different types of transformations as PDA/NDA. The results suggest that transformations that spatially corrupt the image are strong NDA candidates. It can be seen that Random Horizontal Flip is not effective as an NDA; this is because flipping does not spatially corrupt the image but is rather a semantic preserving transformation, hence the NDA distribution $\overline{P}$ is not disjoint from $p_{data}$. On the contrary, it is reasonable to assume that if an image is likely under $p_{data}$, its flipped variant should also be likely. This is confirmed by the effectiveness of this strategy as PDA.

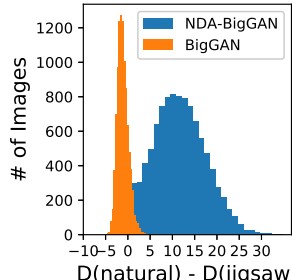

Figure 4: Histogram of difference in the discriminator output for a real image and it's Jigsaw version.

---

[1]We feed a single label to all images to make the architecture suitable for unconditional generation.

Table 1: FID scores over CIFAR-10 using different transformations as PDA and NDA in BigGAN. The results indicate that some transformations yield better results when used as NDA. The common feature of such transformations is they all spatially corrupt the images.

| w/o Aug. | Jigsaw | | Cutout | | Stitch | | Mixup | | Cutmix | | Random Crop | | Random Flip | | Gaussian | |
|---|---|---|---|---|---|---|---|---|---|---|---|---|---|---|---|---|
| | PDA | NDA | PDA | NDA | PDA | NDA | PDA | NDA | PDA | NDA | PDA | NDA | PDA | NDA | PDA | NDA |
| 18.64 | 98.09 | 12.61 | 79.72 | 14.69 | 108.69 | 13.97 | 70.64 | 17.29 | 90.81 | 15.01 | 20.02 | 15.05 | 16.65 | 124.32 | 44.41 | 18.72 |

Table 2: Comparison of FID scores of different types of NDA for unconditional image generation on various datasets. The numbers in bracket represent the corresponding image resolution in pixels. Jigsaw consistently achieves the **best** or second best result.

| | BigGAN | Jigsaw | Stitching | Mixup | Cutout | Cutmix | CR-BigGAN |
|---|---|---|---|---|---|---|---|
| **CIFAR-10 (32)** | 18.64 | **12.61** | 13.97 | 17.29 | 14.69 | 15.01 | 14.56 |
| **CIFAR-100 (32)** | 22.19 | **19.72** | 20.99 | 22.21 | 22.08 | 20.78 | – |
| **CelebA (64)** | 38.14 | 37.24 | **37.17** | 37.51 | 37.39 | 37.46 | – |
| **STL10 (32)** | 26.80 | **23.94** | 26.08 | 24.45 | 24.91 | 25.34 | – |

We believe spatially corrupted negatives perform well as NDA in that they push the discriminator to focus on global features instead of local ones (e.g., texture). We confirm this by plotting the histogram of differences in the discriminator output for a real image and it's Jigsaw version as shown in Fig. 4. We show that the difference is *(a)* centered close to zero for normal BigGAN (so *without NDA training, the discriminator cannot distinguish real and Jigsaw samples well*), and *(b)* centered at a positive number (logit 10) for our method (NDA-BigGAN). Following our findings, in our remaining experiments we use Jigsaw, Cutout, Stitch, Mixup and Cutmix as they achieve significant improvements when used as NDA for unconditional image generation on CIFAR-10.

Table 2 shows the FID scores for BigGAN when trained with five types of negative data augmentation on four different benchmarks. Almost all the NDA augmentations improve the baseline across datasets. For all the datasets except CIFAR-100, $\lambda = 0.25$, whereas for CIFAR-100 it is 0.5. We show the effect of $\lambda$ on CIFAR-10 performance in Appendix H. We additionally performed an experiment using a mixture of augmentation policy. The results (FID 16.24) were better than the baseline method (18.64) but not as good as using a single strategy.

**Conditional Image Generation.** We also investigate the benefits of NDA in conditional image generation using BigGAN. The results are shown in Table 3. In this setting as well, NDA gives a significant boost over the baseline model. We again use $\lambda = 0.25$ for CIFAR-10 and $\lambda = 0.5$ for CIFAR-100. For both unconditional and conditional setups we find the Jigsaw and Stitching augmentations to achieve a better FID score than the other augmentations.

Table 3: FID scores for conditional image generation using different NDAs.[2]

| | BigGAN | Jigsaw | Stitching | Mixup | Cutout | Cutmix | CR-BigGAN |
|---|---|---|---|---|---|---|---|
| **C-10** | 11.51 | **9.42** | 9.47 | 13.87 | 10.52 | 10.3 | 11.48 |
| **C-100** | 15.04 | 14.12 | **13.90** | 15.27 | 14.21 | 13.99 | – |

**Image Translation.** Next, we apply the NDA method to image translation. In particular, we use the Pix2Pix model (Isola et al., 2017) that can perform image-to-image translation using GANs provided paired training data. Here, the generator is conditioned on an image $\mathcal{I}$, and the discriminator takes as input the concatenation of generated/real image and $\mathcal{I}$. We use Pix2Pix for semantic segmentation on Cityscapes dataset (Cordts et al., 2016) (i.e. photos → labels). Table 4 shows the quantitative gains obtained by using Jigsaw NDA[3] while Figure 7 in Appendix F highlights the qualitative improvements. The NDA-Pix2Pix model avoids noisy segmentation on objects including buildings and trees.

---

[2]We use a PyTorch code for BigGAN. The number reported in Brock et al. (2018) for C-10 is 14.73.
[3]We use the official PyTorch implementation and show the best results.

Table 4: Results on CityScapes, using per pixel accuracy (Pp.), per class accuracy (Pc.) and mean Intersection over Union (mIOU). We compare Pix2Pix and its NDA version.

| Metric | Pp. | Pc. | mIOU |
|---|---|---|---|
| Pix2Pix (cGAN) | 0.80 | 0.24 | 0.27 |
| NDA (cGAN) | **0.84** | **0.34** | **0.28** |
| Pix2Pix (L1+cGAN) | 0.72 | 0.23 | 0.18 |
| NDA (L1+cGAN) | **0.75** | **0.28** | **0.22** |

Table 5: AUROC scores for different OOD datasets. OOD-1 contains different datasets, while OOD-2 contains the set of 19 different corruptions in CIFAR-10-C (Hendrycks & Dietterich, 2018) (the average score is reported).

| | | BigGAN | Jigsaw | EBM |
|---|---|---|---|---|
| OOD-1 | DTD | 0.70 | 0.69 | 0.48 |
| | SVHN | 0.75 | 0.61 | 0.63 |
| | Places-365 | 0.35 | 0.58 | 0.68 |
| | TinyImageNet | 0.40 | 0.62 | 0.67 |
| | CIFAR-100 | 0.63 | 0.64 | 0.50 |
| | Average | 0.57 | **0.63** | 0.59 |
| OOD-2 | CIFAR-10-C | 0.56 | **0.63** | 0.60 |

**Anomaly Detection.** As another added benefit of NDA for GANs, we utilize the output scores of the BigGAN discriminator for anomaly detection. We experiment with 2 different types of OOD datasets. The first set consists of SVHN (Netzer et al., 2011), DTD (Cimpoi et al., 2014), Places-365 (Zhou et al., 2017), TinyImageNet, and CIFAR-100 as the OOD datapoints following the protocol in (Du & Mordatch, 2019; Hendrycks et al., 2018). We train BigGAN w/ and w/o Jigsaw NDA on the train set of CIFAR-10 and then use the output value of discriminator to classify the test set of CIFAR-10 (not anomalous) and different OOD datapoints (anomalous) as anomalous or not. We use the AUROC metric as proposed in (Hendrycks & Gimpel, 2016) to evaluate the anomaly detection performance. Table 5 compares the performance of NDA with a likelihood based model (Energy Based Models (EBM (Du & Mordatch, 2019)). Results show that Jigsaw NDA performs much better than baseline BigGAN and other generative models. We did not include other NDAs as Jigsaw achieved the best results.

We consider the extreme corruptions in CIFAR-10-C (Hendrycks & Dietterich, 2018) as the second set of OOD datasets. It consists of 19 different corruptions, each having 5 different levels of severity. We only consider the corruption of highest severity for our experiment, as these constitute a significant shift from the true data distribution. Averaged over all the 19 different corruptions, the AUROC score for the normal BigGAN is **0.56**, whereas the BigGAN trained with Jigsaw NDA achieves **0.63**. The histogram of difference in discriminator's output for clean and OOD samples are shown in Figure 8 in the appendix. High difference values imply that the Jigsaw NDA is better at distinguishing OOD samples than the normal BigGAN.

# 6    REPRESENTATION LEARNING USING CONTRASTIVE LOSS AND NDA

**Unsupervised Learning on Images.** In this section, we perform experiments on three benchmarks: (a) CIFAR10 (C10), (b) CIFAR100 (C100), and (c) ImageNet-100 (Deng et al., 2009) to show the benefits of NDA on representation learning with the contrastive loss function. In our experiments, we use the momentum contrast method (He et al., 2019), *MoCo-V2*, as it is currently the state-of-the-art model on unsupervised learning on ImageNet. For C10 and C100, we train the MoCo-V2 model for unsupervised learning (w/ and w/o NDA) for 1000 epochs. On the other hand, for ImageNet-100, we train the MoCo-V2 model (w/ and w/o NDA) for 200 epochs. Additional hyperparameter details can be found in the appendix. To evaluate the representations, we train a linear classifier on the representations on the same dataset with labels. Table 6 shows the top-1 accuracy of the classifier. We find that across all the three datasets, different NDA approaches outperform *MoCo-V2*. While Cutout NDA performs the best for C10, the best performing NDA for C100 and ImageNet-100 are Jigsaw and Mixup respectively. Figure 9 compares the cosine distance of the representations learned w/ and w/o NDA (jigsaw) and shows that jigsaw and normal images are projected far apart from each other when trained using NDA whereas with original MoCo-v2 they are projected close to each other.

**Transfer Learning for Object Detection.** We transfer the network pre-trained over ImageNet-100 for the task of Pascal-VOC object detection using a Faster R-CNN detector (C4 backbone) Ren et al. (2015). We fine-tune the network on Pascal VOC 2007+2012 trainval set and test it on the 2007 test

Table 6: Top-1 accuracy results on image recognition w/ and w/o NDA on MoCo-V2.

|  | MoCo-V2 | Jigsaw | Stitching | Cutout | Cutmix | Mixup |
|---|---|---|---|---|---|---|
| CIFAR-10 | 91.20 | 91.66 | 91.59 | **92.26** | 91.51 | 91.36 |
| CIFAR-100 | 69.63 | **70.17** | 69.21 | 69.81 | 69.83 | 69.99 |
| ImageNet-100 | 69.41 | 69.95 | 69.54 | 69.77 | 69.61 | **70.01** |

set. The baseline MoCo achieves 38.47 AP, 65.99 AP50, 38.81 AP75 whereas the MoCo trained with mixup NDA gets 38.72 AP, 66.23 AP50, 39.16 AP75 (an improvement of $\approx 0.3$).

**Unsupervised Learning on Videos.** In this section, we investigate the benefits of NDA in self-supervised learning of spatio-temporal embeddings from video, suitable for human action recognition. We apply NDA to Dense Predictive Coding (Han et al., 2019), which is a single stream (RGB only) method for self-supervised representation learning on videos. For videos, we create NDA samples by performing the same transformation on all frames of the video (e.g. the same jigsaw permutation is applied to all the frames of a video). We evaluate the approach by first training the DPC model with NDA on a large-scale dataset (UCF101), and then evaluate the representations by training a supervised action classifier on UCF101 and HMDB51 datasets. As shown in Table 7, Jigsaw and Cutmix NDA improve downstream task accuracy on UCF-101 and HMDB-51, achieving new state-of-the-art performance among single stream (RGB only) methods for self-supervised representation learning (when pre-trained using UCF-101).

Table 7: Top-1 accuracy results on action recognition in videos w/ and w/o NDA in DPC.

|  | DPC | Jigsaw | Stitching | Cutout | Cutmix | Mixup |
|---|---|---|---|---|---|---|
| UCF-101 (Pre-trained on UCF-101) | 61.35 | 64.54 | **66.07** | 64.52 | 63.52 | 63.65 |
| HMDB51 (Pre-trained on UCF-101) | 45.31 | **46.88** | 45.31 | 45.31 | **48.43** | 43.75 |

## 7 RELATED WORK

In several machine learning settings, negative samples are produced from a statistical generative model. Sung et al. (2019) aim to generate negative data using GANs for semi-supervised learning and novelty detection while we are concerned with efficiently creating negative data to improve generative models and self-supervised representation learning. Hanneke et al. (2018) also propose an alternative theoretical framework that relies on access to an oracle which classifies a sample as valid or not, but do not provide any practical implementation. Bose et al. (2018) use adversarial training to generate hard negatives that fool the discriminator for NLP tasks whereas we obtain NDA data from positive data to improve image generation and representation learning. Hou et al. (2018) use a GAN to learn the negative data distribution with the aim of classifying positive-unlabeled (PU) data whereas we do not have access to a mixture data but rather generate negatives by transforming the positive data.

In contrastive unsupervised learning, common negative examples are ones that are assumed to be further than the positive samples semantically. Word2Vec (Mikolov et al., 2013) considers negative samples to be ones from a different context and CPC-based methods (Oord et al., 2018) such as momentum contrast (He et al., 2019), the negative samples are data augmentations from a different image. Our work considers a new aspect of "negative samples" that are neither generated from some model, nor samples from the data distribution. Instead, by applying negative data augmentation (NDA) to existing samples, we are able to incorporate useful inductive biases that might be difficult to capture otherwise (Zhao et al., 2018).

## 8 CONCLUSION

We proposed negative data augmentation as a method to incorporate prior knowledge through out-of-distribution (OOD) samples. NDAs are complementary to traditional data augmentation strategies,

which are typically focused on in-distribution samples. Using the NDA framework, we interpret existing image transformations (e.g., jigsaw) as producing OOD samples and develop new learning algorithms to leverage them. Owing to rigorous mathematical characterization of the NDA assumption, we are able to theoretically analyze their properties. As an example, we bias the generator of a GAN to avoid the support of negative samples, improving results on conditional/unconditional image generation tasks. Finally, we leverage NDA for unsupervised representation learning in images and videos. By integrating NDA into MoCo-v2 and DPC, we improve results on image and action recognition on CIFAR10, CIFAR100, ImageNet-100, UCF-101, and HMDB-51 datasets. Future work include exploring other augmentation strategies as well as NDAs for other modalities.

## 9 ACKNOWLEDGEMENT

The authors would like to thank Shengjia Zhao and Kristy Choi for reviewing an earlier draft of the paper. This research was supprted by NSF (#1651565, #1522054, #1733686), ONR (N00014-19-1-2145), AFOSR (FA9550-19-1-0024), ARO, and Amazon AWS.

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

## A  NUMEROSITY CONTAINMENT

Zhao et al. (2018) systematically investigate generalization in deep generative models using two different datasets: (a) a toy dataset where there are $k$ non-overlapping dots (with random color and location) in the image (see Figure 5a), and (b) the CLEVR dataset where ther are $k$ objects (with random shape, color, location, and size) in the images (see Figure 5b). They train a GAN model (WGAN-GP Gulrajani et al. (2017)) with (either) dataset and observe that the learned distribution does not produce the same number of objects as in the dataset it was trained on. The distribution of the numerosity in the generated images is centered at the numerosity from the dataset, with a slight-bias towards over-estimation. For, example when trained on images with six dots, the generated images contain anywhere from two to eight dots (see Figure 6a). The observation is similar when trained on images with two CLEVR objects. The generated images contain anywhere from one to three dots (see Figure 6b).

In order to remove samples with numerosity different from the train dataset, we use such samples as negative data during training. For example, while training on images with six dots we use images with four, five and seven dots as negative data for the GAN. The resulting distribution of the numerosity in the generated images is constrained to six. We observe similar behaviour when training a GAN with images containing two CLEVR objects as positive data and images with one or three objects as negative data.

## B  IMAGE TRANSFORMATIONS

Given an image of size $H \times W$, the different image transformations that we used are described below.

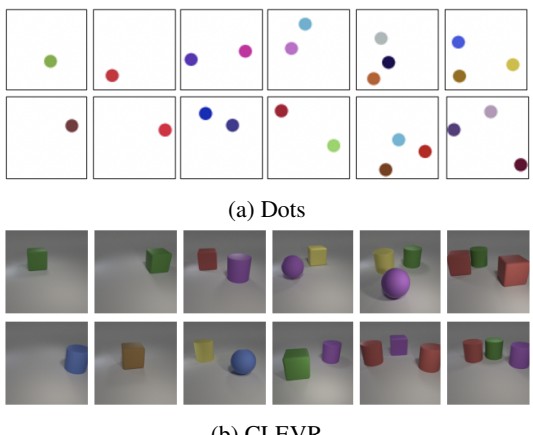

(a) Dots

(b) CLEVR

Figure 5: Toy Datasets used in Numerosity experiments.

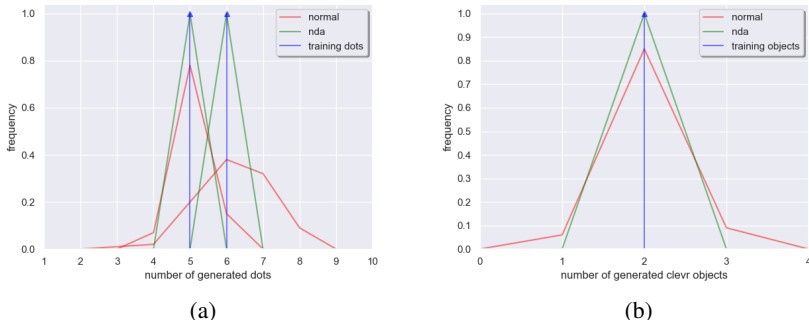

(a)                                                     (b)

Figure 6: **Left**: Distribution over number of dots. The arrows are the number of dots the learning algorithm is trained on, and the solid line is the distribution over the number of dots the model generates. **Right**: Distribution over number of CLEVR objects the model generates. Generating CLEVR is harder so we explore only one, but the behaviour with NDA is similar to dots.

**Jigsaw-$K$** (Noroozi & Favaro, 2016) We partition the image into a grid of $K \times K$ patches of size $(H/K) \times (W/K)$, indexed by $[1, \ldots, K \times K]$. Then we shuffle the image patches according to a random permutation (different from the original order) to produce the NDA image. Empirically, we find $K = 2$ to work the best for Jigsaw-$K$ NDA.

**Stitching** We stitch two equal-sized patches of two different images, either horizontally $(H/2 \times W)$ or vertically $(H \times W/2)$, chosen uniformly at random, to produce the NDA image.

**Cutout / Cutmix** We select a random patch in the image with its height and width lying between one-third and one-half of the image height and width respectively. To construct NDA images, this patch is replaced with the mean pixel value of the patch (like cutout (DeVries & Taylor, 2017) with the only difference that they use zero-masking), or the pixel values of another image at the same location (cutmix (Yun et al., 2019)).

**Mixup-$\alpha$** NDA image is constructed from a linear interpolation between two images $\boldsymbol{x}$ and $\boldsymbol{y}$ (Zhang et al., 2017), $\gamma \boldsymbol{x} + (1 - \gamma)\boldsymbol{y}$; $\gamma \sim \text{Beta}(\alpha, \alpha)$. $\alpha$ is chosen so that the distribution has high density at 0.5.

**Other classes** NDA images are sampled from other classes in the same dataset. See Appendix A.

## C  NDA FOR GANS

**Theorem 1.** *Let $\overline{P} \in \mathcal{P}(\mathcal{X})$ be any distribution over $\mathcal{X}$ with disjoint support than $p_{\text{data}}$, i.e., such that $\text{supp}(p_{\text{data}}) \cap \text{supp}(\overline{P}) = \varnothing$. Let $D_\phi : \mathcal{X} \to \mathbb{R}$ be the set of all discriminators over $\mathcal{X}$,*

$f : \mathbb{R}_{\geq 0} \to \mathbb{R}$ *be a convex, semi-continuous function such that* $f(1) = 0$, $f^{\star}$ *be the convex conjugate of* $f$, $\overline{f}'$ *its derivative, and* $G_{\theta}$ *be a distribution with sample space* $\mathcal{X}$. *Then* $\forall \lambda \in (0, 1]$, *we have:*

$$\underset{G_{\theta} \in \mathcal{P}(\mathcal{X})}{\arg\min} \underset{D_{\phi}:\mathcal{X}\to\mathbb{R}}{\max} L_f(G_{\theta}, D_{\phi}) = \underset{G_{\theta} \in \mathcal{P}(\mathcal{X})}{\arg\min} \underset{D_{\phi}:\mathcal{X}\to\mathbb{R}}{\max} L_f(\lambda G_{\theta} + (1-\lambda)\overline{P}, D_{\phi}) = p_{\text{data}} \quad (4)$$

*where* $L_f(Q, D_{\phi}) = \mathbb{E}_{\boldsymbol{x} \sim p_{\text{data}}}[D_{\phi}(\boldsymbol{x})] - \mathbb{E}_{\boldsymbol{x} \sim Q}[f^{\star}(D_{\phi}(\boldsymbol{x}))]$ *is the objective for* $f$*-GAN (Nowozin et al., 2016). However, the optimal discriminators are different for the two objectives:*

$$\underset{D_{\phi}:\mathcal{X}\to\mathbb{R}}{\arg\max} L_f(G_{\theta}, D_{\phi}) = f'(p_{\text{data}}/G_{\theta}) \quad (5)$$

$$\underset{D_{\phi}:\mathcal{X}\to\mathbb{R}}{\arg\max} L_f(\lambda G_{\theta} + (1-\lambda)\overline{P}, D_{\phi}) = f'(p_{\text{data}}/(\lambda G_{\theta} + (1-\lambda)\overline{P})) \quad (6)$$

*Proof.* Let us use $p(x), \overline{p}(x), q(x)$ to denote the density functions of $p_{\text{data}}, \overline{P}$ and $G_{\theta}$ respectively (and $P$, $\overline{P}$, $Q$ for the respective distributions). First, from Lemma 1 in Nguyen et al. (2008), we have that

$$\underset{D_{\phi}:\mathcal{X}\to\mathbb{R}}{\max} L_f(G_{\theta}, D_{\phi}) = D_f(P\|G_{\theta}) \quad (9)$$

$$\underset{D_{\phi}:\mathcal{X}\to\mathbb{R}}{\max} L_f(\lambda G_{\theta} + (1-\lambda)\overline{P}, D_{\phi}) = D_f(P\|\lambda Q + (1-\lambda)\overline{P}) \quad (10)$$

where $D_f$ refers to the $f$-divergence. Then, we have

$$D_f(P\|\lambda Q + (1-\lambda)\overline{P})$$
$$= \int_{\mathcal{X}} (\lambda q(x) + (1-\lambda)\overline{p}(x)) f\left(\frac{p(x)}{\lambda q(x) + (1-\lambda)\overline{p}(x)}\right)$$
$$= \int_{\mathcal{X}} \lambda q(x) f\left(\frac{p(x)}{\lambda q(x) + (1-\lambda)\overline{p}(x)}\right) + (1-\lambda)f(0)$$
$$\geq \lambda f\left(\int_{\mathcal{X}} q(x) \frac{p(x)}{\lambda q(x) + (1-\lambda)\overline{p}(x)}\right) + (1-\lambda)f(0) \quad (11)$$
$$= \lambda f\left(\frac{1}{\lambda} \int_{\mathcal{X}} \lambda q(x) \frac{p(x)}{\lambda q(x) + (1-\lambda)\overline{p}(x)}\right) + (1-\lambda)f(0)$$
$$= \lambda f\left(\frac{1}{\lambda} \int_{\mathcal{X}} (\lambda q(x) + (1-\lambda)\overline{p}(x) - (1-\lambda)\overline{p}(x)) \frac{p(x)}{\lambda q(x) + (1-\lambda)\overline{p}(x)}\right) + (1-\lambda)f(0)$$
$$= \lambda f\left(\frac{1}{\lambda} - \int_{\mathcal{X}} ((1-\lambda)\overline{p}(x)) \frac{p(x)}{\lambda q(x) + (1-\lambda)\overline{p}(x)}\right) + (1-\lambda)f(0)$$
$$= \lambda f\left(\frac{1}{\lambda}\right) + (1-\lambda)f(0) \quad (12)$$

where we use the fact that $f$ is convex with Jensen's inequality in Eq.(11) and the fact that $p(x)\overline{p}(x) = 0, \forall x \in \mathcal{X}$ in Eq.(12) since $P$ and $\overline{P}$ has disjoint support.

We also have

$$D_f(P\|\lambda P + (1-\lambda)\overline{P}) = \int_{\mathcal{X}} (\lambda p(x) + (1-\lambda)\overline{p}(x)) f\left(\frac{p(x)}{\lambda p(x) + (1-\lambda)\overline{p}(x)}\right)$$
$$= \int_{\mathcal{X}} (\lambda p(x)) f\left(\frac{p(x)}{\lambda p(x) + (1-\lambda)\overline{p}(x)}\right) + (1-\lambda)f(0)$$
$$= \int_{\mathcal{X}} (\lambda p(x)) f\left(\frac{p(x)}{\lambda p(x) + 0}\right) + (1-\lambda)f(0)$$
$$= \lambda f\left(\frac{1}{\lambda}\right) + (1-\lambda)f(0)$$

Therefore, in order for the inequality in Equation 11 to be an equality, we must have that $q(\boldsymbol{x}) = p(\boldsymbol{x})$ for all $\boldsymbol{x} \in \mathcal{X}$. Therefore, the generator distribution recovers the data distribution at the equilibrium posed by the NDA-GAN objective, which is also the case for the original GAN objective.

Moreover, from Lemma 1 in Nguyen et al. (2008), we have that:

$$\arg\max_{D_\phi} L_f(Q, D_\phi) = f'(p_{\text{data}}/Q) \tag{13}$$

Therefore, by replacing $Q$ with $G_\theta$ and $(\lambda G_\theta + (1 - \lambda)\overline{P})$, we have:

$$\arg\max_{D_\phi:\mathcal{X}\to\mathbb{R}} L_f(G_\theta, D_\phi) = f'(p_{\text{data}}/G_\theta) \tag{14}$$

$$\arg\max_{D_\phi:\mathcal{X}\to\mathbb{R}} L_f(\lambda G_\theta + (1 - \lambda)\overline{P}, D_\phi) = f'(p_{\text{data}}/(\lambda G_\theta + (1 - \lambda)\overline{P})) \tag{15}$$

which shows that the optimal discriminators are indeed different for the two objectives. $\qquad\square$

## D   NDA FOR CONTRASTIVE REPRESENTATION LEARNING

We describe the detailed statement of Theorem 2 and proof as follows.

**Theorem 3.** *For some distribution $\overline{p}$ over $\mathcal{X}$ such that $\text{supp}(\overline{p}) \cap \text{supp}(p_{\text{data}}) = \varnothing$, and for any maximizer of the NDA-CPC objective*

$$\hat{h} \in \arg\max_{h_\theta} \max_{g_\phi} \overline{I_{\text{CPC}}}(h_\theta, g_\phi)$$

*the representations of negative samples are disjoint from that of positive samples for $\hat{h}$; i.e., $\forall \boldsymbol{x} \in \text{supp}(p_{\text{data}}), \bar{\boldsymbol{x}} \in \text{supp}(\overline{p})$,*

$$\text{supp}(\hat{h}(\bar{\boldsymbol{x}})) \cap \text{supp}(\hat{h}(\boldsymbol{x})) = \varnothing$$

*Proof.* We use a contradiction argument to establish the proof. For any representation mapping that maximizes the NDA-CPC objective,

$$\hat{h} \in \arg\max_{h_\theta} \max_{g_\phi} \overline{I_{\text{CPC}}}(h_\theta, g_\phi)$$

suppose that the positive and NDA samples share some support, i.e., $\exists \boldsymbol{x} \in \text{supp}(p_{\text{data}}), \bar{\boldsymbol{x}} \in \text{supp}(\overline{p})$,

$$\text{supp}(\hat{h}(\bar{\boldsymbol{x}})) \cap \text{supp}(\hat{h}(\boldsymbol{x})) \neq \varnothing$$

We can always construct $\hat{h}'$ that shares the same representation with $\hat{h}$ for $p_{\text{data}}$ but have disjoint representations for NDA samples; i.e., $\forall \boldsymbol{x} \in \text{supp}(p_{\text{data}}), \bar{\boldsymbol{x}} \in \text{supp}(\overline{p})$, the following two statements are true:

1. $\hat{h}(\boldsymbol{x}) = \hat{h}'(\boldsymbol{x})$;

2. $\text{supp}(\hat{h}'(\bar{\boldsymbol{x}})) \cap \text{supp}(\hat{h}'(\boldsymbol{x})) = \varnothing$.

Our goal is to prove that:

$$\max_{g_\phi} \overline{I_{\text{CPC}}}(\hat{h}', g_\phi) > \max_{g_\phi} \overline{I_{\text{CPC}}}(\hat{h}, g_\phi) \tag{16}$$

which shows a contradiction.

For ease of exposition, let us allow zero values for the output of $g$, and define $0/0 = 0$ (in this case, if $g$ assigns zero to positive values, then the CPC objective becomes $-\infty$, so it cannot be a maximizer to the objective).

Let $\hat{g} \in \arg\max \overline{I_{\text{CPC}}}(\hat{h}, g_\phi)$ be an optimal critic to the representation model $\hat{h_\theta}$ . We then define a following critic function:

$$\hat{g}'(\boldsymbol{x}, \boldsymbol{z}) = \begin{cases} \hat{g}(\boldsymbol{x}, \boldsymbol{z}) & \text{if } \exists \boldsymbol{x} \in \text{supp}(p_{\text{data}}) \quad s.t. \quad \boldsymbol{z} \in \text{supp}(\hat{h}'(\boldsymbol{x})) \\ 0 & \text{otherwise} \end{cases} \tag{17}$$

In other words, the critic assigns the same value for data-representation pairs over the support of $p_{\text{data}}$ and zero otherwise. From the assumption over $\hat{h}$, $\exists \boldsymbol{x} \in \text{supp}(p_{\text{data}}), \bar{\boldsymbol{x}} \in \text{supp}(\overline{p})$, and $\overline{\boldsymbol{z}} \in \text{supp}(\hat{h}(\bar{\boldsymbol{x}}))$,

$$\overline{\boldsymbol{z}} \in \text{supp}(\hat{h}(\boldsymbol{x}))$$

so $(\boldsymbol{x}, \overline{\boldsymbol{z}})$ can be sampled as a positive pair and $\hat{g}(\boldsymbol{x}, \overline{\boldsymbol{z}}) > 0$.

Therefore,

$$\max_{g_\phi} \overline{I_{\text{CPC}}}(\hat{h}', g_\phi) \geq \overline{I_{\text{CPC}}}(\hat{h}', \hat{g}') \tag{18}$$

$$= \mathbb{E}\left[ \log \frac{(n+m)\hat{g}'(\boldsymbol{x}, \boldsymbol{z})}{\hat{g}'(\boldsymbol{x}, \boldsymbol{z}) + \sum_{j=1}^{n-1} \hat{g}'(\boldsymbol{x}, \widehat{\boldsymbol{z}}_j) + \sum_{k=1}^{m} \underbrace{\hat{g}'(\boldsymbol{x}, \overline{\boldsymbol{z}_k})}_{=0}} \right] \quad \text{(plug in definition for NDA-CPC)}$$

$$\geq \mathbb{E}\left[ \log \frac{(n+m)\hat{g}(\boldsymbol{x}, \boldsymbol{z})}{\hat{g}(\boldsymbol{x}, \boldsymbol{z}) + \sum_{j=1}^{n-1} \hat{g}(\boldsymbol{x}, \widehat{\boldsymbol{z}}_j) + \sum_{k=1}^{m} \hat{g}(\boldsymbol{x}, \overline{\boldsymbol{z}_k})} \right] \quad \text{(existence of some}\,\hat{g}(\boldsymbol{x}, \overline{\boldsymbol{z}}) > 0\,)$$

$$= \max_{g_\phi} \overline{I_{\text{CPC}}}(\hat{h}, g_\phi) \quad \text{(Assumption that } \hat{g} \text{ is optimal critic)}$$

which proves the theorem via contradiction. $\qquad\qquad\square$

## E WHAT DOES THE THEORY OVER GANS ENTAIL?

Our goal is to show that NDA GAN objectives are principled in the sense that with infinite computation, data, and modeling capacity, NDA GAN will recover the same optimal generator as a regular GAN. In other words, under these assumptions, NDA will not bias the solution in an undesirable way. We note that the NDA GAN objective is as stable as regular GAN in practice since both methods estimate a lower bound to the divergence with the discriminator, and then minimize that lower bound w.r.t. the generator. The estimated divergences are slightly different, but they have the same minimizer (which is the ground truth data distribution). Intuitively, while GAN and NDA GAN will give the same solution asymptotically, NDA GAN might get there faster (with less data) because it leverages a stronger prior over what the support should (not) be.

## F PIX2PIX

Figure 7 highlights the qualitative improvements when we apply the NDA method to Pix2Pix model (Isola et al., 2017).

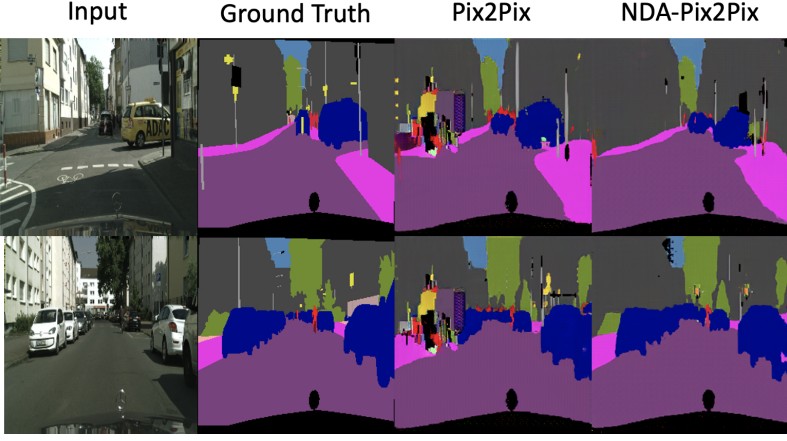

Figure 7: Qualitative results on Cityscapes.

## G  ANOMALY DETECTION

Here, we show the histogram of difference in discriminator's output for clean and OOD samples in Figure 8. High difference values imply that the Jigsaw NDA is better at distinguishing OOD samples than the normal BigGAN.

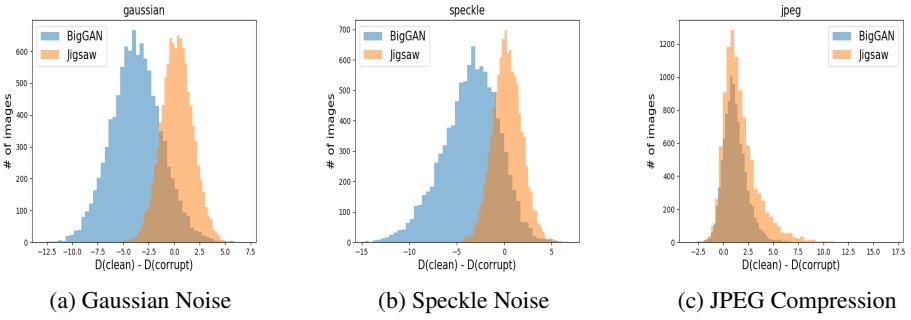

(a) Gaussian Noise      (b) Speckle Noise      (c) JPEG Compression

Figure 8: Histogram of D(clean) - D(corrupt) for 3 different corruptions.

## H  EFFECT OF HYPERPARAMETER ON UNCONDITIONAL IMAGE GENERATION

Here, we show the effect of $\lambda$ for unconditional image generation on CIFAR-10 dataset.

Table 8: Effect of $\lambda$ on the FID score for unconditional image generation on CIFAR-10 using Jigsaw as NDA.

| $\lambda$ | 1.0 | 0.75 | 0.5 | 0.25 | 0.15 |
|---|---|---|---|---|---|
| FID | 18.64 | 16.61 | 14.95 | **12.61** | 13.01 |

## I  UNSUPERVISED LEARNING ON IMAGES

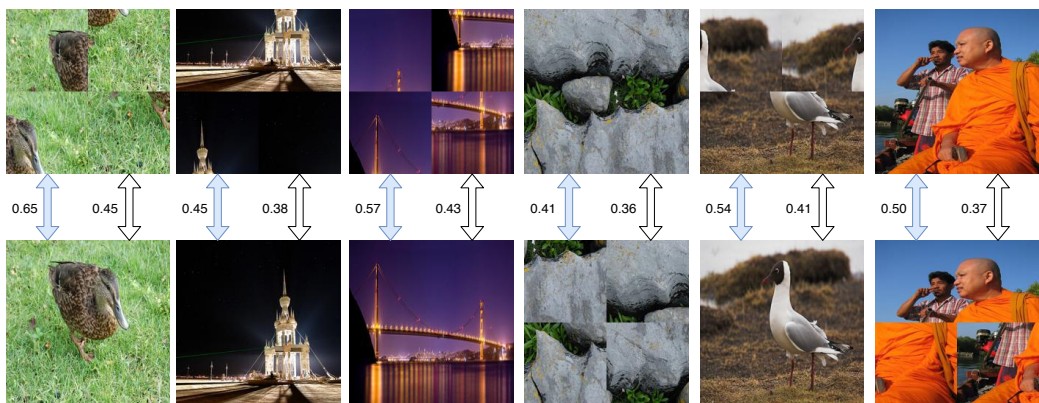

Figure 9: Comparing the cosine distance of the representations learned with Jigsaw NDA and Moco-V2 (**shaded blue**), and original Moco-V2 (**white**). With NDA, we project normal and its jigsaw image representations further away from each other than the one without NDA.

## J  DATASET PREPARATION FOR FID EVALUATION

For dataset preparation, we follow the the following procedures: (a) CIFAR-10 contains 60K $32 \times 32$ images with 10 labels, out of which 50K are used for training and 10K are used for testing, (b)

CIFAR-100 contains 60K $32 \times 32$ images with 100 labels, out of which 50K are used for training and 10K are used for testing, (c) CelebA contains 162,770 train images and 19,962 test images (we resize the images to $64\times64$px), (d) STL-10 contains 100K (unlabeled) train images and 8K (labeled) test images (we resize the images to $32\times32$px). In our experiments the FID is calculated on the test dataset. In particular, we use 10K generated images vs. 10K test images for CIFAR-10, 10K vs. 10K for CIFAR-100, 19,962 vs. 19,962 for CelebA, and 8K vs 8K for STL-10.

## K  Hyperparameters and Network Architecture

**Generative Modeling.**  We use the same network architecture in BigGAN Brock et al. (2018) for our experiments. The code used for our experiments is based over the author's PyTorch code. For CIFAR-10, CIFAR-100, and CelebA we train for 500 epochs whereas for STL-10 we train for 300 epochs. For all the datasets we use the following hyperparameters: batch-size = 64, generator learning rate = 2e-4, discriminator learning rate = 2e-4, discriminator update steps per generator update step = 4. The best model was selected on the basis of FID scores on the test set (as explained above).

**Momentum Contrastive Learning.**  We use the official PyTorch implementation for our experiments. For CIFAR-10 and CIFAR-100, we perform unsupervised pre-training for 1000 epochs and supervised training (linear classifier) for 100 epochs. For Imagenet-100, we perform unsupervised pre-training for 200 epochs and supervised training (linear classifier) for 100 epochs. For CIFAR-10 and CIFAR-100, we use the following hyperparameters during pre-training: batch-size = 256, learning-date = 0.3, temperature = 0.07, feature dimensionality = 2048. For ImageNet-100 pre-training we have the following: batch-size = 128, learning-date = 0.015, temperature = 0.2, feature dimensionality = 128. During linear classification we use a batch size of 256 for all the datasets and learning rate of 10 for CIFAR-10, CIFAR-100, whereas for ImageNet-100 we use learning rate of 30.

**Dense Predictive Coding.**  We use the same network architecture and hyper-parameters in DPC Han et al. (2019) for our experiments and use the official PyTorch implementation. We perform self-supervised training on UCF-101 for 200 epochs and supervised training (action classifier) for 200 epochs on both UCF-101 and HMDB51 datasets.

## L  Code

The code to reproduce our experiments is given here.

## M  Implementation Details

For our experiment over GAN, we augment the batch of real samples with a negative augmentation of the same batch, and we treat the augmented images as fake images for the discriminator. Similarly, for the contrastive learning experiments, we consider negative augmentation of the query image batch as negatives for that batch.

For all our experiments we used existing open-source models. For experiments over GAN, we use the open-source implementations of BigGAN and Pix2Pix models, and for contrastive learning, we use the open-source implementation of the MoCo-v2 model and Dense Predictive Coding. Hence, we did not explain in detail each of the models. Implementing NDA is quite simple as we only need to generate NDA samples from the images in a mini-batch which only takes several lines of code.

## N  Does the gain of NDA for representation learning come from the fact that more negative samples are used?

We perform the experiments over MoCo-v2 which maintains a queue of negative samples. The number of negatives is around 65,536. With our approach, we use the augmented versions of images in the same batch as negative. We transform both the key and query images to create NDA samples.

Thus, the number of negatives for our approach is 65,536 + 2 (one NDA sample created using query image and other using key image), only 0.00003051664 times more than the original number of negatives samples in MoCo-v2. Thus our experiments are comparable to the baseline MoCo-v2. In terms of computation, we need an additional forward pass in each batch to get the representations of the NDA samples. The normal MoCo-v2 requires 1.09 secs for entire forward computation, which includes forward pass through the network, momentum update of the key encoder and dot product between the positive and negative samples. With NDA, 1 forward computation requires 1.36 secs.

## O    WHAT HAPPENS WHEN NEGATIVE DATA AUGMENTATIONS ARE NOISY?

Regarding the performance of negative data augmentation, we perform 2 different experiments:

a) When the noise is low - When using jigsaw as our NDA strategy with a 2 x 2 grid, one out of the 24 permutations will be the original image. We find that when this special permutation is not removed, or there is  4% "noise", the FID score is 12.61, but when it is removed the FID score is 12.59. So, we find that when the noise is low, the performance of our approach is not greatly affected and is robust in such scenarios.

b) When the noise is large - We use random vertical flipping as our NDA strategy, where with 50% probability the image is vertically flipped during NDA. In this case, the "noise" is large, as 50% of the time, the negative sample is actually the original image. We contrast this with the "noise-free" NDA strategy where the NDA image is always vertically flipped. We find that for the random vertical flipping NDA, the FID score of BigGAN is 15.84, whereas, with vertical flipping NDA, the FID score of BigGAN is 14.74. So performance degrades with larger amounts of noise.

