# OpenReview forum: "Negative Data Augmentation "
_ICLR.cc/2021/Conference — ICLR 2021 Poster_

### Official Review · AnonReviewer2 · 2020-10-27
**The idea makes sense but some concerns about the experiment details and theory**

**Rating:** 6
**Confidence:** 4

**Review:**

-- POST REBUTTAL --

The authors addressed well most of my concerns.  I increase my rating. However, the authors need to address all comments of the reviewers and also discuss all missing related works in the updated version.

– Summary –

The paper proposes a new method of leveraging the negative samples (out-of-distribution samples purposely generated from the training data distribution) in generative modeling and representation learning. The main idea aims to leverage the inductive bias of negative samples to constrain the learning of the model, e.g., these negative samples may tell some more information about the support of the data. The experimental results suggest that using these negative samples in GANs (studies with BigGAN model for conditional/unconditional image generation) and contrastive representation learning (study with CPC (Oord et al., 2018) on unsupervised learning on images and videos ) improves the performance of baselines. The paper also reports on improvements in image-image translation and anomaly detection. The paper also provides theorems to prove the convergence of the proposed model on GANs and CPC.

Overall, the paper is easy to read and the idea makes sense. However, I'm a bit concerned about the theory, the significance of improvements and the fairness of the comparison. The paper also misses to discuss and compare with recent works also on data augmentation for GANs.

 -- Strength --

S1 – The usage of negative examples, which obtained from some prior knowledge, to provide the evidence of the learning model about the support/geometry of data distribution sounds reasonable. It has been applied in (Sung et al. (2019)) in semi-supervised learning. The proposed method applies to new applications of generative and representation learnings.

S2 – The experimental results are quite extensive in regards to the applications, and the improvements on GANs quite significant with jigsaw augmentation.

 -- Weakness --

W1 – The paper does not provide the very detailed implementations of proposed models, which is a bit difficult to justify the correctness.

*Generative learning*

W2 – The detail of how to incorporate NDA into GAN is not clear. Also, the PDA baseline for GAN is not detailedly discussed. Does the PDA, NDA, and baseline BigGAN train with the same batch size? I guess that the PDA and NDA had more augmented samples, therefore batch size is larger than the bigGAN baseline?

W3 – The paper does not discuss important related works [a,b,c] of Data Augmentation for GAN recently published. In these papers, they show transforming only real samples (if I understand correctly, it likely is similar to PDA of the proposed baseline) only to train GAN changing the target distribution therefore the generator will learn infrequent or out-of-distribution samples. However, if both real/fake are transformed, data augmentation is helpful in training GAN. Can the authors compare the proposed NDA to at least one of them with the same GAN model?

[a] Differentiable Augmentation for Data-Efficient GAN Training

[b] On Data Augmentation for GAN Training

[c] Training Generative Adversarial Networks with Limited Data


W4 – Eq. 10 showed that $L_f(\lambda * G_{\theta} + (1 - \lambda) * \overline{P}, D_{\phi}) <= D_f(P||\lambda*Q+(1-\lambda * \overline{P}))$, but then infers the lower bound: $D_f(P||\lambda*Q+(1-\lambda * \overline{P})) >= \lambda * f(\frac{1}{\lambda} + (1 - \lambda) f(0)) = D_f(P||\lambda*P+(1-\lambda * \overline{P}))$. Therefore, theoretically I am concerned a bit about the convergence of the model. I wonder whether the authors need an upper bound instead of the lower-bound in this case?

W5 – The paper claimed: “Random Horizontal Flip is not effective as an NDA; this is because flipping does not spatially corrupt the image but is rather a semantic preserving transformation”. How about the Random Vertical Flip only? Can it improve the model since this augmentation looks very good to tell us about the boundary of the data?

*Representation learning*

W6 – The improvements in representation learning do not look significant to me, and the improvements are not too consistent on different datasets according to the type of augmentations.

W7 – The lower bound of Eq. 18 looks just like due to adding a larger batch size for negative samples to train CPC. Can authors compare NDA to CPC with just the same batch size training as the NDA method?

---

> ### Author Response · Authors · 2020-11-22
> **Response to Reviewer 2**
>
> We would like to thank the reviewer for providing insightful ideas and thoughtful comments on the paper. We are pleased to know that the reviewer likes the extensive experiments conducted in the paper, and feels the paper easy to read. We address each of the weakness pointed out by the reviewer
>
> **W1: What are the implementation details?**
>
> **A:**  For our experiment over GAN, we augment the batch of real samples with a negative augmentation of the same batch, and we treat the augmented images as fake images for the discriminator. Similarly, for the contrastive learning experiments, we consider negative augmentation of the query image batch as negatives for that batch.
>
> For all our experiments we used existing open-source models. For experiments over GAN, we use the open-source implementations of BigGAN and Pix2Pix models, and for contrastive learning, we use the open-source implementation of the MoCo-v2 model and Dense Predictive Coding. Hence, we did not explain in detail each of the models.  Implementing NDA is quite simple as we only need to generate NDA samples from the images in a mini-batch which only takes several lines of code. We have also added a link to the code containing our experiments in the Appendix.
>
> **W2: Details of GAN training?**
>
> **A:** Yes, NDA, PDA, and the baseline GAN all use the same batch size, because to generate a negative/positive augmentation we transform the same batch of real images.
>
> **W3: Comparison with some related work.**
>
> **A:** We implemented [b] over the BigGAN model used by us. Since the code is not open-sourced we implemented the paper on our own with Rotation augmentation mentioned in the paper. We get the following results in terms of FID scores over CIFAR-10 :
> Baseline BigGAN - 18.61,
> Our approach - 12.61
> [b] - 15.86
>
> The idea presented in the paper “On Data Augmentation for GAN Training” is complementary to our approach. While the paper proposes to augment images using semantic preserving augmentations and treat them as *positives*, our aim is to use augmentations that *do not preserve the spatial coherence of an image* and use them as *negatives*. Thus, we believe our method is orthogonal to theirs, and the two methods could potentially be even used jointly. In addition, our NDA approach can also be applied to self-supervised learning.
>
> [b] On Data Augmentation for GAN Training
>
> **W4: What does the theory over GANs entail?**
>
> **A:** Our goal is to show that NDA GAN objectives are principled in the sense that with infinite computation, data, and modeling capacity, NDA GAN will recover the same optimal generator as a regular GAN. In other words, under these assumptions, NDA will not bias the solution in an undesirable way. We note that the NDA GAN objective is as stable as regular GAN in practice since both methods estimate a lower bound to the divergence with the discriminator, and then minimize that lower bound w.r.t. the generator. The estimated divergences are slightly different, but they have the same minimizer (which is the ground truth data distribution). Intuitively, while GAN and NDA GAN will give the same solution asymptotically, NDA GAN might get there faster (with less data) because it leverages a stronger prior over what the support should (not) be.
>
> **W5: How about the Random Vertical Flip only?**
>
> **A:** We tried this experiment and we do get an improvement over the normal BigGAN using random vertical flipping. For the CIFAR-10 dataset, the baseline BigGAN FID is 18.61, whereas with random vertical flipping the FID is 15.84, which improves over normal BigGAN but not as much as Jigsaw.
>
> **W6: Consistency and margin of improvement in representation learning.**
>
> **A:** While the improvements might appear small, our approach achieves these improvements with negligible additional costs (only 2 additional negative samples per image in a batch). Across different datasets, we find Jigsaw augmentation to be either performing the best or the second-best, (Jigsaw also worked best for our GAN results). This suggests that Jigsaw is a good candidate for NDA tasks.
>
> For video representation learning tasks, we find that the improvement is significant, an improvement of around 3%, and again Jigsaw is either the best or the second-best performing approach.

---

> > ### Author Response · Authors · 2020-11-22
> > **Response to Reviewer 2 continuation**
> >
> > **W7: Does the gain of NDA come from the fact that more negative samples are used?**
> >
> > **A:** We perform the experiments over MoCo-v2 which maintains a queue of negative samples. The number of negatives is around 65,536. With our approach, we use the augmented versions of images in the same batch as negative. We transform both the key and query images to create NDA samples. Thus, the number of negatives for our approach is 65,536 + 2 (one NDA sample created using query image and other using key image), only 0.00003051664 times more than the original number of negatives samples in MoCo-v2. Thus our experiments are comparable to the baseline MoCo-v2. In terms of computation, we need an additional forward pass in each batch to get the representations of the NDA samples. The normal MoCo-v2 requires 1.09 secs for entire forward computation, which includes forward pass through the network, momentum update of the key encoder and dot product between the positive and negative samples. With NDA, 1 forward computation requires 1.36 secs.

---

### Official Review · AnonReviewer3 · 2020-10-27
**Anonymous review**

**Rating:** 5
**Confidence:** 4

**Review:**

This paper presents a method that uses artificial augmentation of data as Negative (aka OOD) samples to improve various computer vision tasks, including generation, unsupervised learning on images and videos.

Prons:
- The paper is very well written.
- Experiments are comprehensive across different tasks
- The usage of data augmentation seems interesting but with some questions (see below).
- It designs losses for both GANs and contrastive representation learning.
- Code is provided.

Cons:
- Augmentation has been proven in GANs to provide benefits through consistency training  (e.g. CR-GAN, ICLR2020,  Image Augmentations for GAN Training). These samples are used as "positive" samples that should generate consistent predictions. The most famous mixup is also treated as "positive" samples for training. So the augmentation usage here is a bit counterintuitive to me, because you show the opposite conclusion. Is that because only particular augmentation can be used as negative samples, e.g. Jiasaw? The answer to this question is critical. However, the paper does not mention/ study much.
- Advanced self-supervised (contrastive) learning reply on strong augmentation, how negative samples can adapt to these methods? How do we categorize augmentation types used for general cases or NDA cases? Any insights on what kind of augmentations are useful for NDA. For example, in figure 9, the paper proposes to push samples and its jigsaw version away. However, these two pairs share strong local visual contents of objects (just like an image and its crop parts) that usually contrastive learning wants to pull them together. The proposed method tries to push them away. Any insights why it should work?
- If justifications of these questions can be sufficient, I think it can be a strong paper.

---

> ### Author Response · Authors · 2020-11-22
> **Response to Reviewer 3**
>
> We would like to thank the reviewer for providing valuable feedback for the paper. We are pleased that the reviewer finds our approach interesting and the experiments comprehensive.
>
> **Q: Is it important for NDA that particular augmentations are used?**
>
> **A:** Yes. In Section 2, we argued that “NDA strategies are by definition **domain and task specific**”, and in the context of images, we augment the data in ways that preserve the local features but disrupt the global ones. In Table 1, we investigated the augmentation when used as both positive and negative augmentations, where some augmentations such as jigsaw, cutout, mixup work well as negatives and certainly do not work well as positives. [Zhang et al. 2020] also mentioned that **transformations such as cutout do not improve the GAN’s performance when used as positives**, as the generator also starts generating images with cutout artifacts. While mixup is successfully used as positive samples in supervised learning, we note that the same might not be said for tasks such as image generation that we consider.
>
> [Zhang et al. 2020.] Consistency regularization for generative adversarial networks. ICLR 2020
>
> **Q: How can negative samples adapt to methods that use strong augmentations?**
>
> **A:** Our NDA method uses strong augmentations as is, since they are creating positive pairs, and NDA is creating negative pairs.
>
> **Q: How do we categorize augmentations used for general case and NDA case? What kind of augmentations are useful for NDA? Insights on why NDA is useful?**
>
> **A:** Our goal with NDA on self-supervised learning is to learn representations from **semantically inconsistent samples**, such as jigsaw, which preserve the local features but not the global ones. This is in stark contrast to the strong augmentations used in general case self-supervised learning, such as color shift, that **preserve the semantic consistency over the augmented samples, but affect the local features**.  We use some existing augmentation techniques  that introduce semantic inconsistency for creating NDA samples, such as Jigsaw, Cutout, Cutmix, Stitching. Since NDA strategies are domain specific, so the same NDA augmentation techniques might not be useful for all the domains.
>
> As CNN networks are good at learning local features [Geirhos et al. 2019], we find that for a normal MoCo-v2, the representations of an image and its NDA version, such as jigsaw, have high cosine similarity between them even when the two images are not semantically consistent (Figure 9 in our paper). With NDA, we aim to reduce the similarity in representations of two images which share local features but differ in global features. Thus, we want to encourage learning both global and local features from the image.
>
> We believe our approach can be more helpful for learning good representations, since there is evidence that in regular contrastive learning, existence of easy-to learn shared features can suppress learning of other relevant features [Chen and Li, 2020].  In our approach we treat the NDA image, which shares the easy-to-learn features with the query image, as a negative sample. Thus, a network cannot only learn the easy features using our approach.
>
> [Chen and Li, 2020] Intriguing Properties of Contrastive Losses.
>
> [Geirhos et al, 2019] Imagenet-trained cnns are biased towards texture; increasing shape bias improves accuracy and robustness

---

### Official Review · AnonReviewer4 · 2020-10-28

**Rating:** 7
**Confidence:** 4

**Review:**

This paper investigates how augmenting the negative examples, not just the positive examples, can improve a variety of representation learning tasks. The paper investigates a number of different augmentations, and applies them to GANs and contrastive learning with images and videos.

Strengths:
A major strength of the paper is its simplicity. The method is fairly straightforward to implement into several approaches, and it obtains strong results on each approach evaluated in the paper. The approaches evaluated on GANs and contrastive learning with images and videos.

Although the novelty of this method is limited, the paper does a good job at establishing some theoretical results to give intuition why the method works. In contrast to the number of advances in machine learning that lack intuition into why it works, this paper does a good job at offering some explanations and motivations for the approach.

Although this paper focuses on images and videos, the same ideas could be extended to other modalities, such as text or audio, as well.

The experiments are convincing to show the generality of this idea. The experiments are on several different datasets. The experiments are supported by theoretical results, establishing intuition into why the method works. The introduction does a good job at establishing the difference to other data augmentation methods, in particular by using negative examples.

The paper is well written and easy to read.

Weaknesses:
In some cases, the negative data augmentations may actually be inside the positive set. How would the approach scale with noise in the negative augmentations?

---

> ### Author Response · Authors · 2020-11-22
> **Response to Reviewer 4**
>
> We would like to thank the reviewer for providing valuable feedback for the paper. We are pleased that the reviewer likes the simplicity of our work and the good empirical performance of our idea.
>
> We agree that our idea could be expanded to domains other than images and video as well and we plan to explore this in the future.
>
> **Q: What will happen when negative data augmentations are noisy?**
>
> A: Regarding the performance of negative data augmentation, we perform 2 different experiments:
>
> a) When the noise is low - When using jigsaw as our NDA strategy with a 2 x 2 grid, one out of the 24 permutations will be the original image. We find that when this special permutation is not removed, or there is ~4% “noise”, the FID score is 12.61, but when it is removed the FID score is 12.59. So, we find that when the noise is low, the performance of our approach is not greatly affected and is robust in such scenarios.
>
> b) When the noise is large - We use random vertical flipping as our NDA strategy, where with 50% probability the image is vertically flipped during NDA. In this case, the “noise” is large, as 50% of the time, the negative sample is actually the original image. We contrast this with the “noise-free” NDA strategy where the NDA image is always vertically flipped. We find that for the random vertical flipping NDA, the FID score of BigGAN is 15.84, whereas, with vertical flipping NDA, the FID score of BigGAN is 14.74.  So performance degrades with larger amounts of noise.

---

### Official Review · AnonReviewer1 · 2020-10-28
**Review for Negative Data Augmentation**

**Rating:** 9
**Confidence:** 4

**Review:**

1) Summary
- The authors proposed the negative data augmentation technique which is useful for generative adversarial networks, anomaly detection, self-supervised learning frameworks.
- The idea is simple, and the technique was proven that it is powerful for several tasks.
- They performed several experiments, and I think the experiments were enough to show the technique's superiority.

2) Strong points
- Good idea
- Strong experimental results
- Simple to use
- Easy to understand

3) Weak points
- In Figure 3, they claim that in the absence of NDA. the support of a generative model learned from samples may "over-generalize"...
- I am not sure that the sentence is true.

This paper is well written, and concrete, I recommend that this paper should be presented in ICLR 2021.

---

> ### Author Response · Authors · 2020-11-22
> **Response to Reviewer 1**
>
> We would like to thank the reviewer for positive comments on our work. We are encouraged that the reviewer found our idea easy to use, and one which achieves strong empirical results.
>
> **Q: What does “over-generalize” mean in the context?**
>
> A: Indeed, characterizing what constitutes the correct generalization behavior for a generative model is not easy when all that we observe is finite data ([Zhao et al. 2018]). That’s actually one of the motivations of this work, as we provide a way to control “what not to generate” with negative samples.
>
> In Figure 3 we show a generative model that when trained on images containing only 6 objects, not only generates images with 6 objects, but also images with 2 or 8 objects (even though there are no such images in the training set). We argue this could be “over-generalization” -- if the user really wanted to learn a distribution over images with a varying number of objects, they would probably have used them in the training set. With our NDA approach, we can explicitly use images with 2 or 8 objects as negative images to avoid such “over-generalization”.
>
> Note that a certain amount of generalization is needed for the model to be useful -- we certainly don’t want it to just memorize the training set. However, while some generalizations are desirable, some clearly aren’t. NDA provides some level of control over this to the user.
>
> [Zhao et al. 2018] Bias and generalization in deep generative models. NeurIPS 2018

---

### Decision · Program_Chairs · 2021-01-07
**Final Decision**

**Decision:**

Accept (Poster)

**Comment:**

All reviewers find the proposed data augmentation approach simple, interesting and effective. They agree that paper does a good job exploring this idea with number of experiments. However the paper also suffers from some drawbacks, and reviewers raise questions about some of the conclusions of the paper - in particular how to designate an augmentation as either negative or positive is not clear apriori to training. While I agree with this criticism, I believe the paper overall explores an interesting direction and provides a good set of experiments than can be built on in  future works, and I suggest acceptance. I encourage authors to address all the reviewers concerns as per the feedback in the final version.